# General strategy for developing thick-film micro-thermoelectric coolers from material fabrication to device integration

Xiaowen Sun[1,2,3], Yuedong Yan [2,3] ✉, Man Kang[1,2], Weiyun Zhao[2], Kaifen Yan[2], He Wang[2], Ranran Li[2], Shijie Zhao[2], Xiaoshe Hua[2], Boyi Wang[2], Weifeng Zhang [2] ✉ & Yuan Deng [1,2] ✉

Micro-thermoelectric coolers are emerging as a promising solution for high-density cooling applications in confined spaces. Unlike thin-film micro-thermoelectric coolers with high cooling flux at the expense of cooling temperature difference due to very short thermoelectric legs, thick-film micro-thermoelectric coolers can achieve better comprehensive cooling performance. However, they still face significant challenges in both material preparation and device integration. Herein, we propose a design strategy which combines $Bi_2Te_3$-based thick film prepared by powder direct molding with micro-thermoelectric cooler integrated via phase-change batch transfer. Accurate thickness control and relatively high thermoelectric performance can be achieved for the thick film, and the high-density-integrated thick-film micro-thermoelectric cooler exhibits excellent performance with maximum cooling temperature difference of 40.6 K and maximum cooling flux of 56.5 W·cm$^{-2}$ at room temperature. The micro-thermoelectric cooler also shows high temperature control accuracy (0.01 K) and reliability (over 30000 cooling cycles). Moreover, the device demonstrates remarkable capacity in power generation with normalized power density up to 214.0 μW · cm$^{-2}$ · K$^{-2}$. This study provides a general and scalable route for developing high-performance thick-film micro-thermoelectric cooler, benefiting widespread applications in thermal management of microsystems.

As the integration level and heat flux of various microsystems (such as 5 G communication chips, laser chips, etc.) increase, their performance increasingly depends on efficient thermal management[1]. However, conventional cooling systems based on compression or caloric effects are not suitable for microsystems, due to their large volume or extreme operation conditions[2,3]. Thermoelectric coolers (TECs) based on the Peltier effect provide an alternative technology directly generating local cooling by electricity, which has many advantages such as no moving parts, no noise, lightweight, and long service life[1,2,4]. Especially, micro-TECs (μ-TECs) are capable of achieving high cooling power density to meet the rising demand for high-power heat dissipation (>40 W cm$^{-2}$)[5–7] and precise temperature control (<0.1 K)[1,2] in microsystems. The maximum cooling power density ($q_{cmax}$) of a TEC is defined as the cooling flux when the temperature difference across the

[1]School of Materials Science and Engineering, Beihang University, Beijing, China. [2]Key Laboratory of Intelligent Sensing Materials and Chip Integration Technology of Zhejiang Province, Hangzhou Innovation Institute of Beihang University, Hangzhou, China. [3]These authors contributed equally: Xiaowen Sun, Yuedong Yan ✉ e-mail: yuedongyan@buaa.edu.cn; zhangweifeng@buaa.edu.cn; dengyuan@buaa.edu.cn

device is zero. Neglecting the contact resistances, it can be expressed by the following formula[6,8]:

$$q_{cmax} = \frac{\left(S_p - S_n\right)^2 T_c^2}{4\left(\frac{1}{\sigma p} + \frac{1}{\sigma n}\right)} \cdot \frac{f}{H} \tag{1}$$

where $S$ and $\sigma$ are the Seebeck coefficient and electrical conductivity of thermoelectric (TE) legs, respectively, the subscripts p and n represent the carrier type of TE legs, $f$ is the filling factor defined as the ratio of total leg cross-area to device area, $H$ and $T_c$ denote the height of TE legs and the cold-side temperature of device, respectively. It follows that $q_{cmax}$ can be effectively enhanced by improving the performance of TE materials, increasing the filling factor, and reducing the height of TE legs.

$Bi_2Te_3$-based materials are among the best TE materials near room temperature (RT = 300 K) and are currently the optimal choice for constructing μ-TECs for thermal management of microsystems[8–13]. However, the long TE legs (>100 μm) and low filling factor (usually <50%) of commercial $Bi_2Te_3$-based μ-TECs result in a low cooling power density of typically less than 30 W cm$^{-2}$, which restricts their applications in microsystem cooling[14,15]. Using thin films (<10 μm) as TE legs can potentially enhance the cooling power density, but it also substantially reduces the cooling temperature difference across the μ-TEC owing to the effect of inevitable contact resistances between the TE legs and electrodes[16,17]. Although some studies have reported high-performance μ-TECs fabricated with single crystal $Bi_2Te_3$-based superlattice thin films, their high material and fabrication costs hinder their commercial feasibility[6,18]. Therefore, $Bi_2Te_3$-based TE thick films (10–100 μm) are more suitable for μ-TECs to attain higher cooling power density while preserving high cooling temperature difference.

However, the preparation of high-performance $Bi_2Te_3$-based TE thick films and the high-density integration of μ-TECs both remain challenging. Conventional mechanical cutting is difficult to prepare $Bi_2Te_3$-based thick films due to the intrinsic brittleness of materials. Thin-film growth methods, such as physical vapor deposition[19,20], chemical vapor deposition[6,21], or electrochemical deposition[5,22,23] are time-consuming and inefficient for thick-film fabrication. Currently, slurry printing is the most effective method for obtaining $Bi_2Te_3$-based thick films[24–30]. However, this method is complex and environmentally unfriendly due to the use of various organic ingredients (solvent, binder, and dispersant). These ingredients also cause high porosity and low mechanical strength of the sintered TE thick films[27,28]. Furthermore, the integration of high-performance μ-TECs encounters several issues[24–26,28–30]: the fluidity of TE slurry reduces printing accuracy and hinders the high-density integration of the device; the high surface roughness and oxidation raise the contact resistances between the TE thick films and electrodes; it is also hard to produce the diffusion barrier on both sides of the thick-film legs and transfer the legs onto the electrodes. To date, only a few studies have managed to fabricate TECs based on slurry-printed TE thick films, and the highest reported cooling temperature difference of the TECs is only 6.2 K[25,29,30]. Consequently, there is still an urgent need to put forward a strategy of fabricating $Bi_2Te_3$-based thick films with high TE and mechanical properties and integrating high-performance μ-TECs.

In this study, we present a strategy based on powder direct molding and phase-change batch transfer to overcome challenges from material fabrication to device integration, thereby developing high-performance $Bi_2Te_3$-based thick-film μ-TECs. Instead of complex slurry printing, we introduce a simple pre-pressing mold to directly perform the TE powder, achieving the rapid preparation of $Bi_2Te_3$-based thick films with accurate thickness control. In addition, vacuum sealing technology is employed to improve the TE performance of thick films by inhibiting oxidation during post-annealing. The prepared TE thick films have high performance in power factor (PF = $S^2\sigma$), the

figure of merit (ZT = $S^2\sigma T/\kappa$, $T$, and $\kappa$ are the Kelvin temperature and thermal conductivity of TE materials, respectively), and mechanical strength. Moreover, we innovatively propose a batch transfer method based on phase-change materials to achieve the high-density integration of μ-TEC, which can result in a high filling factor of over 64%. The integrated thick-film μ-TEC exhibits an excellent comprehensive cooling performance with a maximum cooling temperature difference of 40.6 K and a maximum cooling flux of 56.5 W cm$^{-2}$ at RT, which significantly exceeds that of previously reported other polycrystalline film-based μ-TECs. The μ-TEC also shows a high-temperature control accuracy of 0.01 K and reliability of over 30,000 cooling cycles. Additionally, the μ-TEC demonstrates an impressive power generation performance with a normalized power density of 214.0 μW cm$^{-2}$ K$^{-2}$, setting a record among reported polycrystalline $Bi_2Te_3$-based TE devices. Such remarkable cooling and power generation performance can be attributed to the achievement of high-performance TE thick films and high-density device integration methods.

## Results
### Fabrication and characterization of $Bi_2Te_3$-based thick films
Figure 1a presents a schematic illustration of the powder direct molding method, which is employed to achieve the rapid fabrication of high-performance $Bi_2Te_3$-based TE thick films without introducing various organic components, complex process flows, and expensive equipment. A sophisticated homemade pre-pressing mold with a shallow sink was used to directly preform $Bi_2Te_3$-based TE powder into thick films, which were then densified by applying a high mechanical pressure of 800 MPa. After the compacted thick films were transferred to a small quartz tube, the vacuum sealing and annealing process (VS-A) was conducted. Consequently, the batch preparation of high-performance $Bi_2Te_3$-based TE thick films can be realized (Fig. 1b). In this simple fabrication approach, the pre-pressing mold can precisely control the thickness of TE films by spreading the powder evenly in the sink. Since the shallow depth of the pre-pressing mold is only 100–200 μm, it is crucial to select appropriate powder particle size and mold surface roughness to ensure even powder distribution. Figure 1c shows the scanning electron microscope (SEM) images of the cross-sections of $Bi_2Te_3$ thick films with varying thicknesses, displaying great thickness uniformity and surface flatness. In addition, the introduction of vacuum sealing technology is also an important step in obtaining high-performance TE thick films. The oxygen content and TE properties of $Bi_2Te_3$ thick films under different annealing conditions have been compared as illustrated in Fig. 1d and Supplementary Fig. 1. Unannealed $Bi_2Te_3$ thick films contain approximately 4% oxygen (O) by atomic percentage due to the oxidation of TE powder in the air. Traditional annealing in a large tube furnace filled with inert gas (Ar) atmosphere leads to further oxidation of thick films by residual oxygen, resulting in a low power factor of 24.08 μW cm$^{-1}$ K$^{-2}$ for $Bi_2Te_3$ thick films (treated at 400 °C for 90 min). In contrast, VS-A thick films (treated at 440 °C for 90 min) exhibit only slightly higher oxygen content than that of unannealed samples, indicating the effectiveness of high vacuum sealing in preventing oxidation during annealing. Moreover, confining the samples in a small sealed space may reduce the volatilization of elements to a certain extent. As a result, the VS-A process can achieve a high power factor of 33.6 μW cm$^{-1}$ K$^{-2}$, which is significantly higher than that of traditionally annealed thick films (Supplementary Fig. 2).

To optimize the TE performance, vacuum-sealed $Bi_2Te_3$ thick films were subjected to heat treatment at various annealing temperatures and durations. The structure and composition of the annealed TE thick films were first characterized and analyzed. The SEM images of the sample surface and cross-section show that all the TE thick films obtained under different annealing conditions have a high compactness (Fig. 2a and Supplementary Fig. 3), which is crucial for achieving the high mechanical strength of TE materials. Indeed, Fig. 2b

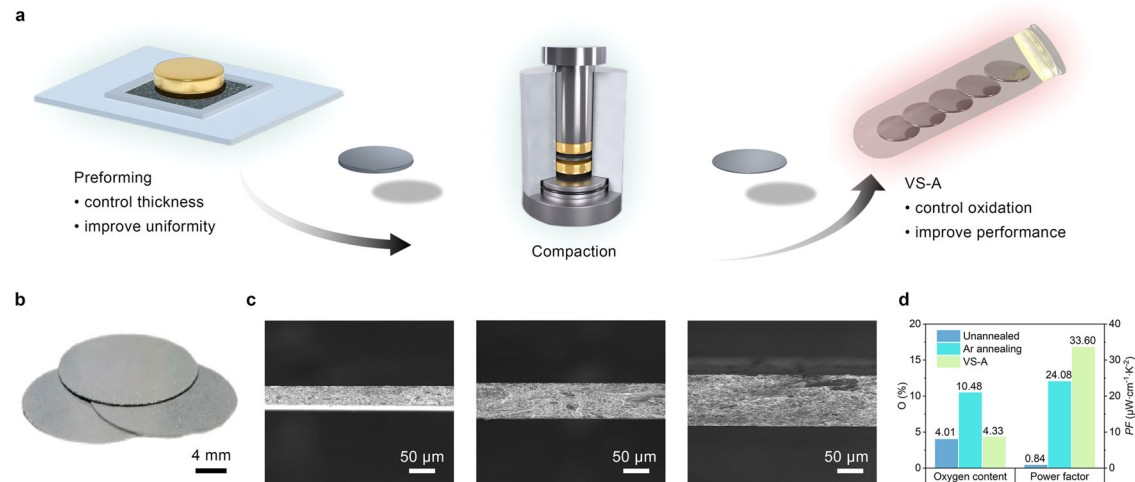

**Fig. 1 | Fabrication of Bi₂Te₃-based TE thick films. a** Schematic illustration of the powder direct molding method consisting of preforming, compaction, and vacuum seal-annealing (VS-A) processes. **b** Digital photo of the fabricated Bi₂Te₃ TE thick films. **c** SEM images of the cross-sections of Bi₂Te₃ thick films with thicknesses of 40, 75, and 106 μm. **d** Comparison of the Bi₂Te₃ thick films obtained under different annealing conditions (unannealed, traditional Ar annealing in a tube furnace, and VS-A) on the oxygen (O) content and RT power factor (*PF*).

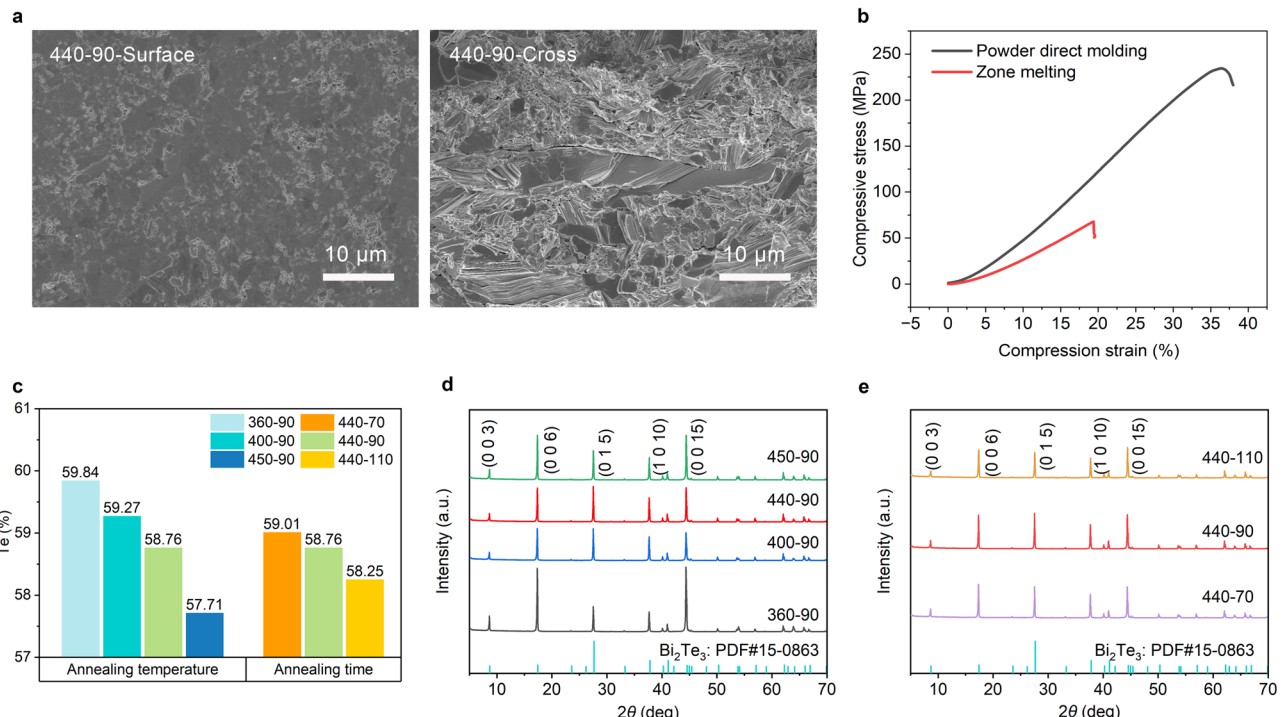

**Fig. 2 | Characterization of structure and composition of Bi₂Te₃ thick films prepared by the powder direct molding method. a** SEM images of the surface and cross-sectional microstructure of samples annealed at 440 °C for 90 min. **b** Compressive strength of Bi₂Te₃ as a function of compression strain. The powder direct molding and commercial zone melting samples are compared. **c** Atomic percentage of Te content in Bi₂Te₃ thick films prepared at different annealing temperatures (360, 400, 440, and 450 °C) and times (70, 90, and 110 min), characterized by EDS. **d** XRD spectra of Bi₂Te₃ thick films annealed at different annealing temperatures for 90 min. **e** XRD spectra of Bi₂Te₃ thick films annealed at 440 °C for different times.

demonstrates that the compressive strength of Bi₂Te₃ prepared by the powder direct molding method is about 3.7 times that of the commercial zone melting samples (see Methods for detailed mechanical property characterization). Such high mechanical strength can ensure that the integrated TE devices have high impact resistance. Figure 2c and Supplementary Table 1 display the energy-dispersive X-ray spectroscopy (EDS) characterization of annealed Bi₂Te₃ thick films, revealing that the volatilization of Te is still inevitable even though the

annealing was performed in a confined space. The atomic percentage of Te content gradually decreases with increasing annealing temperature and duration. Notably, the Te content in the samples annealed at 450 °C for 90 min (450-90) has a significant decrease compared to the 440-90 samples. This may be due to the proximity of the annealing temperature of 450 °C to the melting point of Te, which accelerates the volatilization of Te. To understand the effect of annealing conditions on crystallinity, we conducted X-ray diffraction

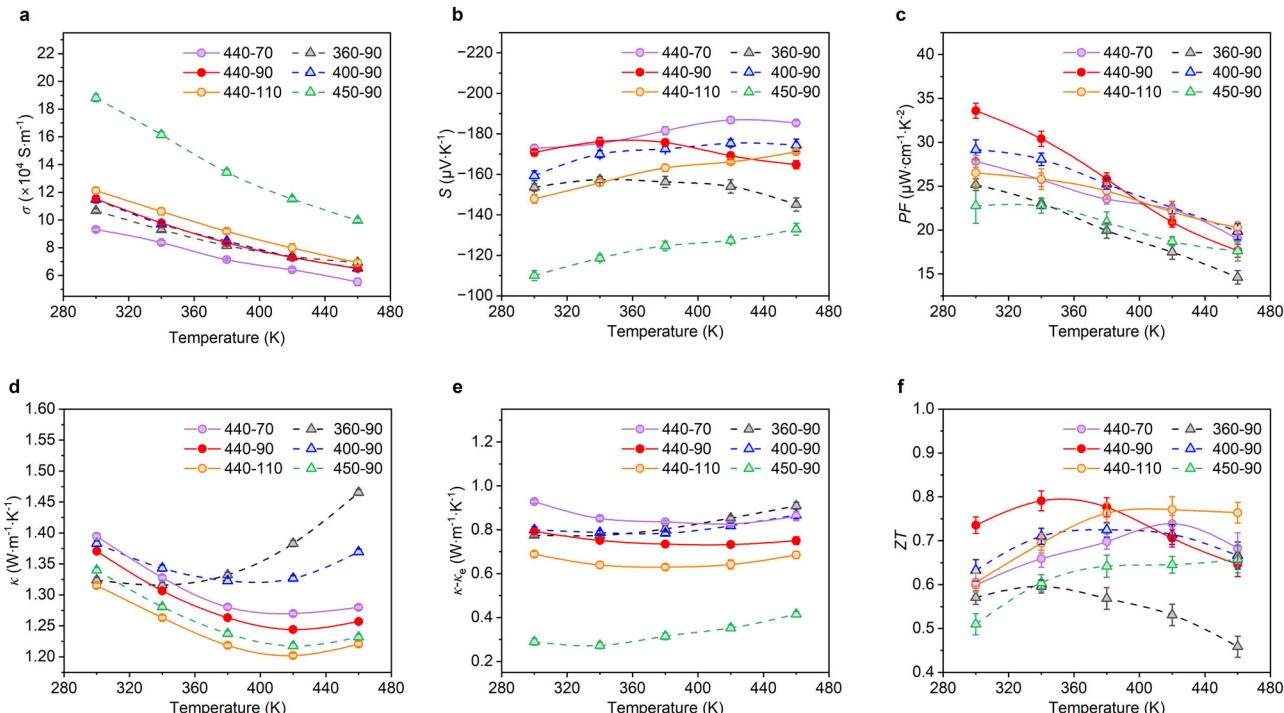

**Fig. 3 | Temperature-dependent in-plane TE properties of Bi₂Te₃ thick films prepared under different annealing conditions. a** Electrical conductivity ($\sigma$). **b** Seebeck coefficient ($S$). **c** Power factor ($PF$). **d** Total thermal conductivity ($\kappa$). **e** Lattice thermal conductivity ($\kappa-\kappa_e$). **f** TE figure of merit ($ZT$). Data shown are mean ± S.E.M. of three independent measurements ($n=3$).

(XRD) characterization. Figure 2d, e shows that the intensity of the (0 1 5) main peak of Bi₂Te₃ thick films initially increases with annealing temperature and time, suggesting improved crystallinity. However, the main peak intensity is then reduced under higher annealing temperature (450-90) and longer annealing time (440-110), which can be attributed to the excessive volatilization of Te, as characterized by EDS (Fig. 2c).

Furthermore, the temperature-dependent in-plane TE properties of Bi₂Te₃ thick films obtained under different annealing conditions were measured. Figure 3a shows that the TE thick films exhibit a typical metallic electrical conductivity-temperature dependence, and the RT conductivity increases with annealing temperature and time. Based on the analyses of EDS and XRD characterizations, annealing-induced crystallinity improvement can reduce carrier scattering, while volatilization-induced Te vacancy defects can provide more free electrons[31-33]. Therefore, both the crystallinity improvement and Te volatilization during the annealing process will enhance the electrical conductivity of Bi₂Te₃ thick films. The substantial increase in electrical conductivity of the 450-90 samples can be attributed to the excessive volatilization of Te, as demonstrated by the EDS characterization (Fig. 2c). Figure 3b displays that the absolute RT Seebeck coefficient first increases and then decreases with annealing temperature, while monotonically decreases with annealing time. This can be understood as that during the annealing process, the Seebeck coefficient initially increases with the enhancement of crystallinity, but then decreases as the carrier concentration increases due to the volatilization of Te. As a result, a maximum RT power factor of $33.6\,\mu\text{W}\,\text{cm}^{-1}\,\text{K}^{-2}$ can be achieved for the 440-90 samples, as shown in Fig. 3c. In addition, the thermal conductivity of samples was also measured, presenting a non-monotonic temperature dependence that first decreases and then increases (Fig. 3d), which is in line with the bipolar diffusion mechanism[8]. The RT thermal conductivity has a similar dependence on annealing temperature and time as the Seebeck coefficient, and it can also be explained by crystallinity and Te volatilization. The thermal

conductivity initially increases as the crystallinity is improved by annealing. However, further annealing-induced Te vacancy defects lead to enhanced phonon scattering and thus reduce the lattice thermal conductivity, which is demonstrated by subtracting electronic thermal conductivity from the total thermal conductivity (Fig. 3e and Supplementary Fig. 4). Consequently, the trade-off between electrical conductivity, Seebeck coefficient, and thermal conductivity leads to an optimized RT $ZT$ value of about 0.73 in the Bi₂Te₃ thick films annealed at 440 °C for 90 min (Fig. 3f). To analyze the anisotropy of materials, the RT out-of-plane Seebeck coefficient, thermal conductivity, and electrical conductivity of the prepared N-type Bi₂Te₃ thick film annealed at 440 °C for 90 min were also characterized using the portable Seebeck tester, laser flash method, and Cox–Strack method (Supplementary Fig. 5 and Note 1), respectively. These results are displayed in Supplementary Table 2. By comparing the RT in-plane TE properties (Fig. 3), it can be found that the electrical conductivity and thermal conductivity show a clear anisotropy, while the Seebeck coefficient and $ZT$ do not exhibit significant anisotropy. Predictably, the powder direct molding method can achieve higher TE performance of Bi₂Te₃-based thick films by eliminating powder oxidation[34,35], regulating element doping[36,37], introducing nanostructures[13,28], and so on.

## Integration of thick-film μ-TECs

A typical cross-plane μ-TEC mainly consists of ceramic substrates, electrodes, and TE legs, forming electrical serial and thermal parallel connections of TE legs. To achieve comprehensive performance with high cooling temperature difference and high cooling power density, we established a simplified one-dimensional model of the TEC containing a pair of TE legs (Supplementary Fig. 6), and determined the optimized TE film thickness and device filling factor through calculation and analysis (Supplementary Note 2)[14]. Bi₂Te₃ and Bi₀.₅Sb₁.₅Te₃ prepared by the powder direct molding method were used as N-type and P-type TE materials for TEC, respectively. We adopted the out-of-

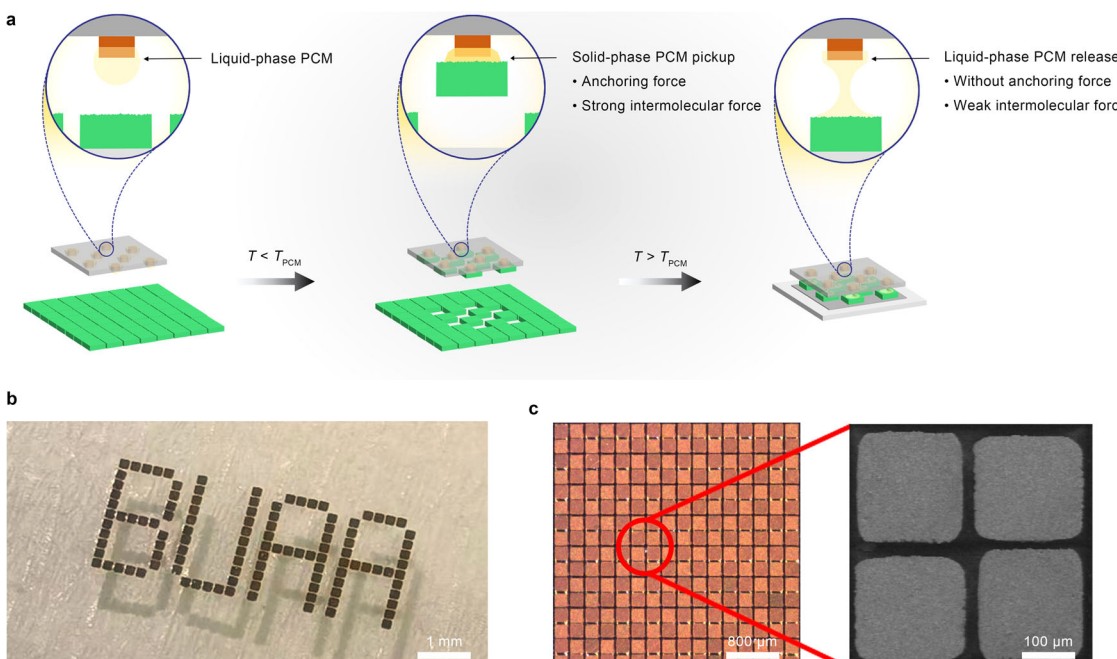

**Fig. 4 | Selective batch transfer based on phase change materials (PCMs).**
**a** Schematic illustration of phase-change batch transfer. The cyan square array represents micro-components. **b** Digital photo of a "BUAA" pattern formed by batch transfer of TE legs with a size of 200 × 200 μm². **c** Optical and SEM images of a high-density array with P-type and N-type TE legs arranged alternately via phase-change batch transfer. The rosin is chosen as the phase change material.

plane TE properties (Supplementary Table 2) for simulation calculations, assuming that the material properties are temperature-independent. Based on these material parameters, the current-dependent cooling temperature difference ($\Delta T$) and power density ($q_c$) were calculated using the one-dimensional model and finite element method (FEM) simulations (Supplementary Fig. 7). The results of these two methods are consistent, demonstrating the reliability of the one-dimensional model. Based on the one-dimensional model, the dependence of the cooling performance of TEC on the TE leg height and device filling factor was calculated by considering reasonable electrical and thermal contact resistances between the TE legs and electrodes (Supplementary Figs. 8 and 9). On the one hand, a high cooling power density can be reached with a relatively small TE leg height, whereas the maximum cooling temperature difference rapidly decreases when the TE leg height is less than 100 μm. On the other hand, increasing the filling factor can greatly enhance the cooling power density with little effect on the cooling temperature difference. Therefore, it is necessary to choose an appropriate thickness (~100 μm) of TE thick films and the highest possible filling factor to integrate μ-TECs with comprehensive cooling performance.

In order to rapidly integrate thick-film TE materials into μ-TECs with a high filling factor, a key issue that needs to be addressed is the high-density and high-yield transfer of a large number of micro-TE legs. Batch transfer technology is desirable to be employed, because transferring TE legs one by one is of low efficiency, poor uniformity, and high alignment accuracy requirements. However, the state-of-the-art batch transfer technologies based on vacuum adsorption, electromagnetic force, van der Waals force, laser irradiation, sticky glue, or self-assembly remain plagued with some drawbacks[38–40], including difficulties in transferring high-density micro-component array, low transfer yield due to poor adhesion control, surface pollution of the components, complex and expensive equipment, and inability to transfer non-planar components. Herein, we come up with a batch transfer strategy based on phase change materials (PCMs). The schematic diagram of the route is depicted in Fig. 4a. A stamp with convex points is coated by a low-melting-point PCM, which can undergo a

reversible solid–liquid phase transition in response to temperature changes. After attaching and infiltrating the liquid-phase PCM to the surface of (planar or non-planar) micro-components, the temperature is controlled below the phase-change temperature ($T_{PCM}$) to solidify the PCM. The solid-phase PCM can not only provide a strong inter-molecular force but also form a strong anchoring force at the PCM-micro-component interface due to the surface roughness of the micro-components. This adhesion force allows the convex points to selectively pick up micro-components. When the picked micro-components are attached to the receiving substrate, the temperature is raised above $T_{PCM}$ to melt the PCM. The liquid-phase PCM has a weak inter-molecular force, and the interfacial anchoring force disappears. The greatly weakened adhesion force enables easy release of micro-components from the stamp. As a result, the strength of adhesion can be significantly modulated by manipulating the intermolecular force and anchoring force, achieving batch transfer with a high success rate. The selection of a suitable PCM for batch transfer of TE legs involves several critical requirements: low melting point, significant adhesion changes during phase transition, and ease of cleaning. In our study, we evaluated various low melting point phase change materials, including ice, paraffin wax, polyethylene glycol, and rosin. Finally, the ethanol-soluble rosin with a proper melting point of 110 °C is chosen due to its highest success rate for the clean transfer of TE legs. Based on this phase-change batch transfer method, micro-components with any size can be arranged into any pattern by designing the size and distribution of convex points on the stamp. Figure 4b presents the "BUAA" pattern formed by batch transfer of TE legs with a size of 200 × 200 μm². In addition, non-destructive and non-polluting high-density batch transfer of micro TE legs can be achieved using the rosin as the PCM. Figure 4c and Supplementary Fig. 10 show that the P-type and N-type TE legs can be alternately transferred precisely to form a high-density array, with a gap of less than 50 μm between adjacent TE legs and no PCM residue after ethanol cleaning. Thus, this batch transfer method has great potential in the high-density array integration of various micro-components including micro-TE elements, micro-sensors, and micro-LEDs.

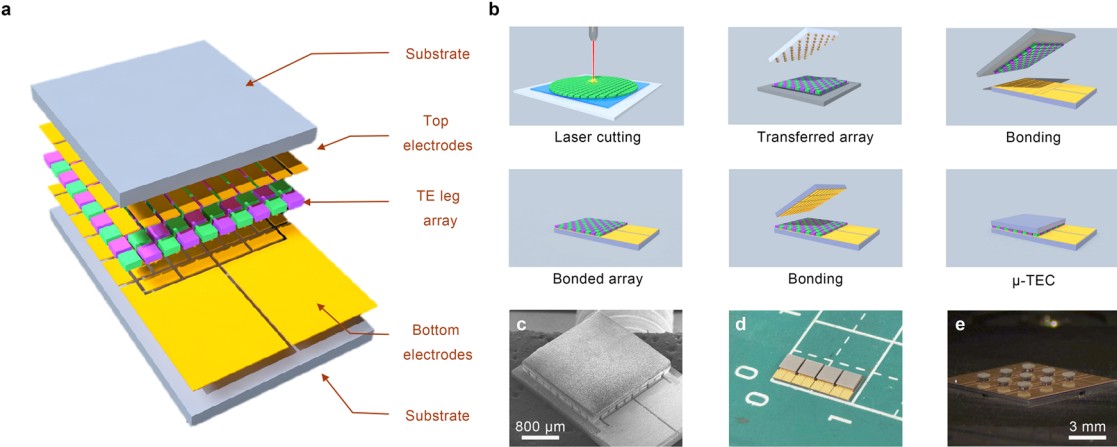

**Fig. 5 | Integration and images of thick-film μ-TECs. a** Composition and structure of a thick-film μ-TEC with a high filling factor. **b** Device integration process of the μ-TEC. The femtosecond laser, phase-change batch transfer, and alignment bonding technology technologies are employed. **c, d** SEM image and digital photo of as-integrated thick-film μ-TECs with a device thickness of only 0.6 mm and a top surface size of 2.5 × 2.5 mm². **e** Optical image of multiple μ-TECs integrated on a single substrate at once. One μ-TEC has a small top surface size of 0.7 × 0.7 mm² and contains two pairs of TE legs with a dimension of 250 × 250 × 100 μm³.

Based on the aforementioned strategy, μ-TEC with both micro-TE legs and high filling factor can be integrated successfully (Fig. 5a). Figure 5b presents the integration process of thick-film μ-TECs. After depositing Ni/Au diffusion barrier layers on both sides of the thick films (Supplementary Fig. 11), the femtosecond laser technology was adopted to cut 100 μm thick N-type $Bi_2Te_3$ and P-type $Bi_{0.5}Sb_{1.5}Te_3$ TE films into TE legs with a size of 200 × 200 μm². Due to the high precision of laser processing, the gap between adjacent TE legs can be less than 50 μm (Supplementary Fig. 12). These micro-TE legs were then selectively transferred to a receiving substrate using the phase-change batch transfer method to form a high-density TE leg array with P–N alternating arrangement. After that, the transferred TE leg array and the upper and lower electrodes were integrated into a μ-TEC by the soldering procedure. Figure 5c, d shows the preparation of μ-TECs that have a device thickness of only 0.6 mm and an internal resistance of about 3.5 Ω. After subtracting the resistance of the TE legs and electrodes from the total internal resistance, the electrical contact resistivity at the TE leg-electrode interface can be estimated to be about $1.02 \times 10^{-10}$ Ω m². This low contact resistivity is consistent with that obtained by the Cox–Strack measurements (Supplementary Note. 1). Additionally, the μ-TEC contains 50 pairs of TE legs within a range of 2.5 × 2.5 mm² (top surface size), indicating a high filling factor of approximately 64%. If the processing accuracy from the TE thick films to the TE legs is improved, a higher filling factor can be achieved through the high-density batch transfer method. Moreover, Fig. 5e demonstrates that this integration process can realize a high-density one-time integration of multiple μ-TECs on a single substrate whose integration procedures are demonstrated in Supplementary Fig. 13, providing a potential application for high-resolution array cooling or temperature sensing[41,42].

**Cooling and power generation performance of thick-film μ-TECs**
Aiming to demonstrate the performance advantages of the thick-film μ-TECs integrated in this work, the cooling and power generation capacity of the μ-TEC with a size of 2.5 × 2.5 mm² (Fig. 5c) were systematically studied. An infrared thermal imager was used to measure the dependence of the cold-side temperature of the μ-TEC on the input current, while a 25 °C cold plate was used to dissipate heat from the hot side. The maximum cooling temperature difference ($\Delta T_{max}$, defined as the cold plate temperature minus the lowest device cold-side temperature) of 40.6 K was obtained with a current of about 1.1 A ($I_{max}$) applied. Based on the known material parameters (Supplementary Table 2) and electrical contact resistivity (Supplementary Fig. 5), the

experimental data were analyzed using FEM (Fig. 6a, b). The current-dependent $\Delta T$ can be well fitted by the FEM simulation, and a low thermal contact resistivity of about $4 \times 10^{-6}$ m² K W$^{-1}$ at the TE leg-electrode interface can be extracted. Further simulation was performed to calculate the dependence of $q_c$ and performance coefficient (COP) on input current, and thus the maximum values of $q_c$ and COP at $I_{max}$ can be obtained as 56.5 W cm$^{-2}$ ($q_{cmax}$) and 0.7 (COP$_{max}$), respectively (Supplementary Fig. 14). An experimental characterization was also performed, and $q_{cmax}$ of 55.4 W/cm² and COP$_{max}$ of 0.66 at $I_{max}$ and 25 °C were obtained. The measured results are slightly smaller than the simulated values, which can be attributed to the effect of parasitic thermal resistance and measurement error in the experiment. Figure 6c illustrates that the device exhibits a fast cooling response of 4.8 ms K$^{-1}$ at 1.1 A when the cold-side temperature drops from 25 °C to below 0 °C. Notably, the cooling response of our μ-TEC is much faster than the commercial mini-TEC and bulk-TEC (the inset in Fig. 6c), which can be attributed to the higher $q_{cmax}$. In addition, the temperature control precision and cooling cycling reliability of the device have also been demonstrated. For the temperature control measurements, a thermistor was used to record the cold-side temperature precisely, providing feedback for a current control module with PID to regulate the cold-side temperature precisely. Figure 6d and Supplementary Fig. 15 show that the cold-side temperature can be accurately controlled with a high precision of less than 0.01 K over a wide temperature range, suggesting a great potential for applications in optical chips[1,2]. Figure 6e presents that the cooling performance has hardly degraded after more than 30,000 cycles of rapid cooling between 25 °C and −12 °C, demonstrating the robustness of the thick-film μ-TEC. Finally, compared with the reported polycrystalline film-based μ-TECs, the as-integrated thick-film μ-TEC exhibits higher comprehensive cooling performance in terms of $\Delta T_{max}$ and $q_{cmax}$ (Fig. 6f)[5,17,43–47].

The power generation performance was also measured with various temperature differences applied between the cold side and hot side of the μ-TEC. Figure 6g plots the output current and power as a function of output voltage under different temperature differences. The extracted open-circuit voltage ($V_{oc}$) and maximum output power ($P_{max}$) exhibit linear and parabolic dependencies with respect to the temperature difference, respectively. The results agree with the theoretical predictions (Supplementary Fig. 16). By fitting the temperature difference dependence of $V_{oc}$, an effective Seebeck coefficient of 277 μV K$^{-1}$ for one pair of TE legs can be obtained, which is roughly 72% of the theoretical value (385 μV K$^{-1}$). This is mainly because the temperature difference across the TE legs is only a fraction of the total

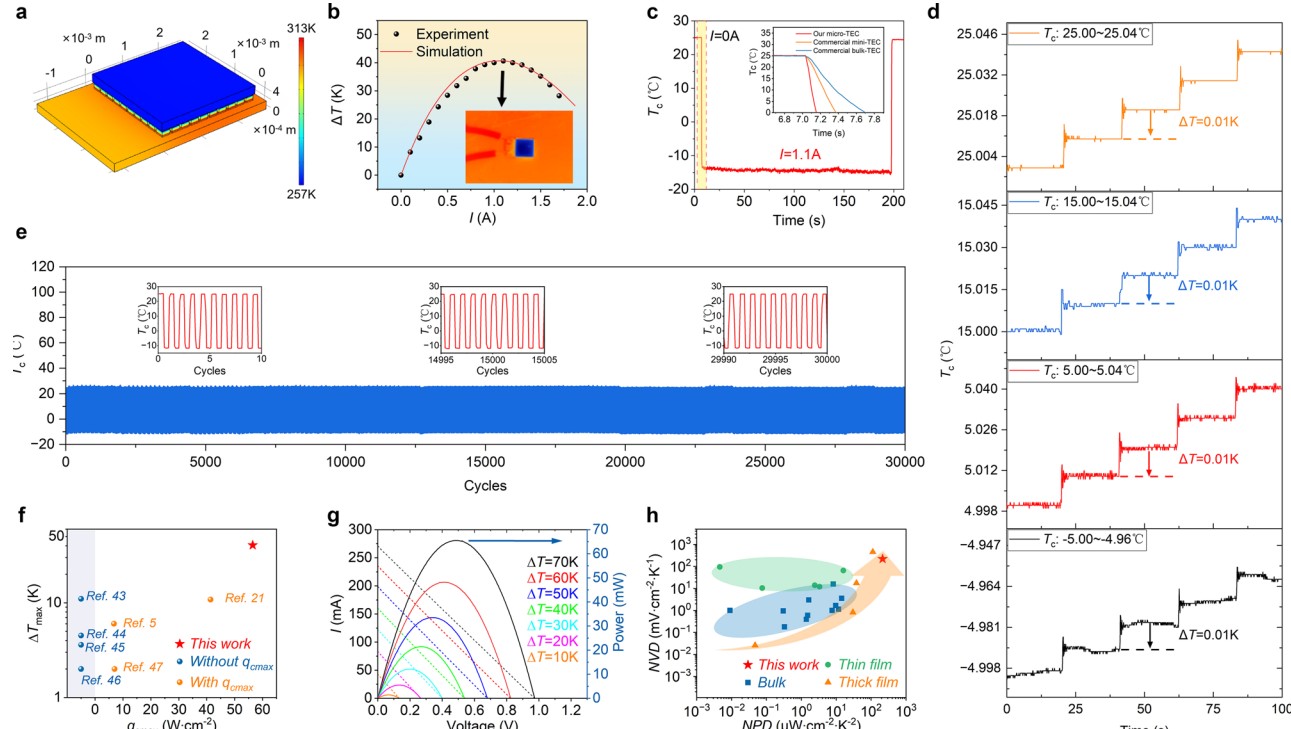

**Fig. 6 | Cooling and power generation performance of the as-integrated thick-film μ-TECs containing 50 pairs of TE legs. a** FEM simulation of device cooling performance using COMSOL Multiphysics® software. **b** Cooling temperature difference as a function of input electric current. The black dots represent experimental data measured by infrared thermal imaging at 25 °C, and the red line represents the FEM results. The lower right inset shows the infrared thermal image with an applied current of 1.1 A. **c** Transient response of device cold-side temperature ($T_c$) when a current of 1.1 A is applied at 25 °C. The inset shows the comparison of our μ-TEC and the commercial mini-TEC and bulk-TEC on transient response. **d** Precise control of $T_c$ over a wide temperature range from −5 to 25 °C.

When the temperature is adjusted every 20 s in a step of 0.01 K, clear discernible temperature changes can be observed. **e** Cooling cycling reliability of the μ-TEC under a continuous on-off pulse current of 0.9 A, represented as the $T_c$ as a function of cooling cycles. **f** Comparison of the as-integrated thick-film μ-TECs and reported other polycrystalline $Bi_2Te_3$-based thick-film and thin-film μ-TECs on $\Delta T_{max}$ and $q_{cmax}$ [5,17,43–47]. **g** Output current (dashed lines) and power (solid lines) as functions of the output voltage at different temperature differences applied between the hot side and cold side of the device. **h** Comparison of the as-integrated thick-film μ-TECs and reported polycrystalline $Bi_2Te_3$-based TE devices on maximum normalized voltage density (NVD) and normalized power density (NPD).

temperature difference, indicating that the $V_{oc}$ can be further improved by reducing the parasitic thermal resistances[7]. In order to compare power generation performance with the reported TE devices, the maximum normalized voltage density (NVD = $V_{oc}/(\Delta T \cdot A_D)$) and normalized power density (NPD = $P_{max}/(\Delta T^2 \cdot A_D)$) were calculated, where $A_D$ denotes the device area. Figure 6h demonstrates that the thick-film μ-TEC in this work has the highest NPD (214.0 μW cm⁻² K⁻²) among the reported polycrystalline $Bi_2Te_3$-based TE devices and a very high NVD (222.6 mV cm⁻² K⁻¹). The high power generation and cooling performance exhibited in our thick-film μ-TEC can be attributed to the use of high-performance TE thick films with optimized thickness and the device integration route that can achieve a high filling factor.

## Discussion

In summary, the high-performance $Bi_2Te_3$-based thick-film μ-TEC has been fabricated through powder direct molding and phase-change batch transfer. The rapid preparation and accurate thickness control of $Bi_2Te_3$-based TE thick film can be achieved by directly pre-forming powder with a pre-pressing mold. The vacuum sealing process is able to effectively enhance the TE performance of thick film. By controlling annealing conditions to balance crystallinity and elemental volatilization, an optimized RT $ZT$ value is obtained. Furthermore, easy-to-clean phase-change materials are used to realize batch transfer of high-density TE leg arrays by manipulating the anchoring force and intermolecular force. This method enables the large-scale integration of thick-film μ-TECs with a high filling factor. Besides, it also provides a powerful tool for the high-density array integration of various micro-

components, such as micro-sensors and micro-LEDs. The as-prepared μ-TEC shows excellent comprehensive performance in cooling temperature difference (40.6 K), cooling power density (56.5 W cm⁻²), temperature control accuracy (0.01 K), and cycling reliability (30,000 cycles). It is worth noting that a remarkable power generation performance of the thick-film μ-TEC has also been demonstrated. The normalized power density and normalized voltage density reach high values of 214.0 μW cm⁻² K⁻² and 222.6 mV cm⁻² K⁻¹, respectively. This study unlocks a general, facile, and scalable strategy for developing chip-level high-performance μ-TECs, which can be widely applied in microelectronic cooling, optical communication temperature control, biomedical device temperature control, and even micro-power generation.

## Methods

### Preparation of $Bi_2Te_3$-based TE thick films

A powder direct molding method was proposed to prepare $Bi_2Te_3$-based TE thick films. $Bi_2Te_3$ and $Sb_{1.5}Bi_{0.5}Te_3$ powders with a purity of 99.99% and a diameter of 10−30 μm were purchased from Kaiyada (Hangzhou, China) and Dongxin Mill (Wuhan, China), respectively. The TE powders were directly prepared into thick films in several simple steps. First, a homemade pre-pressing mold was fabricated using stainless steel sheets and $Al_2O_3$ ceramic plates. Specifically, a stainless steel sheet with a thickness of 50−150 μm was cut into a required pattern using a femtosecond laser cutter (PHAROS-20, Amplitude Laser Group). The wavelength, pulse frequency, pulse width, and power of the laser are 343 nm, 100 kHz, 290 fs and 6 W, respectively.

The operation is conducted at ambient temperature. The 50 μm thick polyimide adhesive was then used to paste the patterned stainless steel sheet onto a $50 \times 100 \times 1$ mm$^3$ Al$_2$O$_3$ ceramic plate with an appropriate surface roughness of 200–300 nm, forming a shallow sink with a size of $30 \times 30$ mm$^2$ and a depth of 100–200 μm. After evenly distributing TE powders into the sink of pre-pressing mold, a punch was used to apply a pressure of 150 MPa for 3 s on the powders to obtain pre-formed thick films with a diameter of 20 mm. Subsequently, the thick films were loaded into a high-pressure mold and subjected to a second compression at 800 MPa for 20 s. Finally, the compacted thick films were vacuum sealed in a small quartz tube with a vacuum level of $10^{-2}$ Pa, followed by a post-annealing at different temperatures (360, 400, 440, and 450 °C) for different durations (70, 90, and 110 min) in a tube furnace.

### Characterization of TE thick films

The surface and cross-sectional microstructure of the TE thick films were characterized using SEM (Sigma 300, Zeiss). The element content and crystal structure were analyzed by EDS (X-act SDD, Oxford Instruments) and XRD (Empyrean-100, Malvern Panalytical), respectively. The in-plane electrical conductivity and Seebeck coefficient were simultaneously measured in a helium atmosphere from 300 to 460 K using a TE testing system (CTA-3S, Beijing Cryoall Science and Technology Co., Ltd.). Based on the electrical conductivity and Seebeck coefficient, the electronic thermal conductivity was calculated as shown in Supplementary Fig. 4. The out-of-plane Seebeck coefficient was measured by a portable Seebeck tester (PTM-3, Wuhan Jiayitong Technology Co., Ltd). The out-of-plane electrical conductivity was measured using the Cox-Strack method (Supplementary Note 1). The thermal conductivity ($\kappa$) was calculated by the formula $\kappa = \rho \cdot D \cdot C_p$, where $\rho$, $D$, and $C_p$ are the density, thermal diffusivity, and specific heat capacity, respectively. The $\rho$ was obtained to be 7.54 g/cm$^3$ for Bi$_2$Te$_3$ and 6.60 g/cm$^3$ for Sb$_{1.5}$Bi$_{0.5}$Te$_3$ by measuring the mass and geometrical dimensions of the thick films, the $D$ was measured by the laser flash method (LFA-467, NETZSCH), and the typical $C_p$ values of 159 J/(kg K) and 200 J/(kg K) were used for Bi$_2$Te$_3$ and Sb$_{1.5}$Bi$_{0.5}$Te$_3$, respectively. It is difficult to directly measure the compressive strength of the prepared films because its thickness is only less than 200 μm. Therefore, to obtain the mechanical properties, a thicker Bi$_2$Te$_3$ sample with a thickness of 2 mm and a diameter of 10 mm was fabricated using the powder direct molding method under the same preparation condition. As a comparison, the commercial zone melting Bi$_2$Te$_3$ sample was cut into the same size as the powder direct molding sample. Then, the compressive strain-stress curves of these Bi$_2$Te$_3$ samples were measured by an electro-mechanical universal testing machine (RGM-6300, Shenzhen Reger Instrument Co., Ltd.).

### Integration of thick-film μ-TECs

The Bi$_2$Te$_3$ and Sb$_{1.5}$Bi$_{0.5}$Te$_3$ thick films with a thickness of 100 μm were prepared as the N-type and P-type TE materials to integrate μ-TECs. First, 1 μm thick Ni and 200 nm thick Au layers were sequentially deposited on both sides of the TE thick film by magnetron sputtering after an in-situ plasma cleaning. The Ni and Au serve as diffusion barrier and anti-oxidation layers, respectively. After that, the surface-metallized thick films were attached to the polydimethylsiloxane (PDMS) films, and then a femtosecond laser cutter was employed to cut the thick films into TE legs with a dimension of $200 \times 200$ μm$^2$. The gap between adjacent TE legs is less than 50 μm due to the high laser processing precision. Subsequently, 50 μm thick Cu on an aluminum nitride (AlN) substrate was etched to form a stamp with Cu convex points (Supplementary Fig. 10). After coating the rosin on convex points, this stamp was used to transfer TE legs on a receiving PDMS substrate by the phase-change batch transfer method detailed in the main text. Finally, the transferred TE leg array with P-type and N-type legs arranged alternately was

bonded to the upper and lower electrodes using a solder, and thus the μ-TEC was fabricated. The electrodes mainly consist of 25 μm Cu, 5 μm Ni, and 200 nm Au electroplated on the AlN substrate with a thickness of 200 μm. Similarly, the phase-change batch transfer method can also achieve the integration of multiple μ-TECs on a single substrate at once. According to the electrode positions where TE legs need to be welded, corresponding convex point stamps were prepared (Supplementary Fig. 13). After coating rosin on the convex point surfaces, P-type and N-type TE legs were picked up and alternately transferred to the PDMS substrate, followed by soldering with upper and lower electrodes.

### Performance characterization of thick-film μ-TECs

The internal resistance of the μ-TEC was determined by the standard four-terminal measurements. To measure the cooling performance, the hot side of the μ-TEC was attached to a cold plate using the thermal grease, and the temperature of the cold plate was controlled at 25 °C by a commercial bulk TEC module to provide stable heat dissipation. Then, a DC power supply was used to provide current to the μ-TEC, while the cold-side temperature of the device was measured by an infrared thermal imager (FTIR, Thermo Scientific Nicolet iS20) or a K-type thermocouple (TT-K, Omega). The temperature measurement accuracy of the infrared thermal imager was calibrated by the K-type thermocouple. The transient cooling response of commercial mini-TEC and bulk-TEC (Wangu Co., Ltd., Quzhou, China) at optimal current was also characterized in the same measurement conditions. For the cooling cycling measurements, an on-off pulsed current was continuously applied by a Keithley 2400 digital multimeter, and the pulse time and amplitude were set to be 10 seconds (1:1 on/off ratio) and 0.9 A, respectively. For the precise temperature control measurements, a thermistor (DT103F3930, EXSENSE Sensor Technology) was used to record the cold-side temperature, providing feedback for a current control module with PID to precisely regulate the cold-side temperature. In addition, the experimental characterization of $q_{cmax}$ and COP$_{max}$ of our μ-TEC was performed in Ferrotec Co., Ltd. To characterize the power generation performance, the μ-TEC was sandwiched between the cold and hot plates using the thermal grease for interface contact. The temperature of the hot plate was adjusted between 30 °C and 90 °C while the temperature of the cold plate was maintained at 20 °C, thereby establishing a temperature difference across the μ-TEC to generate electrical output. The Keithley 2400 was used to record the output voltage and current.

### Simulation of cooling performance

To fabricate μ-TECs with high comprehensive cooling performance, a simplified one-dimensional model was used to optimize the TE leg height and device filling factor. This model equates the TEC to simple one-dimensional thermal and electrical circuits (Supplementary Fig. 6), and then assumes that the TEC operates in an adiabatic environment with ideal heat dissipation of hot side. While taking into account the actual material parameters and reasonable contact resistances, the cooling performance under steady-state conditions was calculated as a function of the TE leg height or filling factor. The detailed discussions about the theoretical simulations are presented in Supplementary Note 2. After integrating the μ-TEC and measuring its current-dependent cooling temperature difference, FEM simulations based on multi-physics field coupling were performed using a COMSOL software package, to accurately fit the experimental data (Fig. 6b) and further calculate the cooling power density and performance coefficient (Supplementary Fig. 14).

### Reporting summary

Further information on research design is available in the Nature Portfolio Reporting Summary linked to this article.

## Data availability

Source data are provided in this paper. Additional data related to this work are available from the corresponding authors upon request. Source data are provided in this paper.

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

## Acknowledgements

The authors acknowledge financial support from the National Key Research and Development Program of China (grant No. 2018YFA0702100 (Y.D.)), the Zhejiang Provincial Key Research and Development Program of China (grant No. 2021C01026 (Y.D.)), the Zhejiang Provincial Natural Science Foundation of China (grant No. LZ23E020004 (WF.Z.)), the National Natural Science Foundation of China (grant No. 52003015 (WF.Z.)) and the Leading Innovative and Entrepreneur Team Introduction Program of Zhejiang (2020R01007 (Y.D.)). The authors are grateful for Ming Li in Ferrotec Co., Ltd. for the helpful characterization of $q_c$.

## Author contributions

Y.Y., W.F.Z., and Y.D. designed and supervised the work. X.S., Y.Y., and M.K. prepared the thick-film thermoelectric materials and characterized the material properties. X.S., Y.Y., W.Y.Z., H.W., R.L., S.Z., X.H., and W.F.Z. fabricated the integrated μ-TECs and carried out the device performance characterization. Y.Y., K.Y., and B.W. performed the theoretical simulations and analytical calculations. X.S., Y.Y., W.F.Z., and Y.D. analyzed the data and wrote the paper.

## Competing interests

The authors declare no competing interests.
