## [Peer Review File · Nature Communications]

General strategy for developing thick-film micro-thermoelectric coolers from material fabrication to device integrationREVIEWER COMMENTS

Reviewer #1 (Remarks to the Author):

In this manuscript, the authors prepared a thick-film micro-thermoelectric cooler through powder direct molding route and batch transfer strategy, which exhibits good cooling performance, precise temperature control and high normalized power density. The micro-TEC shows great potential in thermal management or fast cooling of microsystems. Since the design strategy is proposed for the first time to the best of our knowledge, the idea of this work is original and novel. Besides, the article is well written and the experiments are also conducted in detail. I think both the material fabrication method and the integration strategy could provide guidelines for the researchers in thermoelectric and device micromachining fields. Thus, I recommend the acceptance of the manuscript after addressing the following minor issues.

1. In the process of fabricating thermoelectric thick films, the authors employed a homemade pre-pressing mold to control the thickness and uniformity of the films. I think it is necessary to provide a detailed description about how the pre-pressing mold is prepared in the section of "Methods".
2. The authors propose a novel and interesting selective batch transfer method based on the phase transition material to achieve the integration of micro-TECs with high-density thermoelectric leg arrays. In this method, it is better to explain why rosin is chosen as the phase change material instead of other materials.
3. In Fig. 5e, nine μ -TECs with a size of $0.7 \times 0.7 \text{ mm}^2$ were integrated on a single substrate. How is the one-time integration process of multiple devices achieved?
4. In the section of cooling performance characterization, the authors mention that "Figure 6c illustrates that the device exhibits a fast cooling response of 4.8ms/K ". To make it easy for readers to inspect, the transient response time when changing the current should also be clearly presented in Fig. 6c.

Reviewer #2 (Remarks to the Author):

The rapidly increased heat flux for the integrated electronics results in a daunting challenge for highly efficient heat dissipation. Designing solid-state thermoelectric cooling devices with large cooling power density is of great significance for electronic cooling/thermal management applications. Herein, Sun et al. developed thick-film thermoelectric coolers with an excellent cooling flux of 56 W cm^{-2} . In addition, the micro-thermoelectric devices also show high-temperature control accuracy (0.01 K) and reliability (over 30000 cooling cycles). The results are highly meaningful for the development of advanced micro-thermoelectric devices. Therefore, I would like to recommend it to be accepted by Nature Communications. My suggestions for further improving the manuscript are provided below for consideration

1. The cold-side temperature of the micro-TEC is obtained from the infrared thermal imager. However, since the error of the thermal imager could be large, if it is accurate to determine the temperature precision as high as 0.01 K ? In addition, the Q_c and COP are based on the simulation results, if it is possible to experimentally characterize these cooling performances.
2. The anisotropy in the thermoelectric properties of n-type Bi_2Te_3 -based alloys is very noticeable, I am wondering if the thermoelectric properties of the prepared thick films are obtained from the same direction. It is understandable that the electrical properties in the out-of-plane direction and thermal conductivity in the in-plane direction are difficult to characterize for the film. Comments on the anisotropy should be added to the revised

manuscript.

3. Since the thickness of the prepared films is only less than 200 micrometers, are the compressive strain and stress results directly obtained from the film sample? Or if the results are obtained from the micropillar sample? Details for the mechanical properties characterization should be provided.

4. The BiTeSe alloys have improved thermoelectric performance compared to the Bi₂Te₃. Please comment on why Bi₂Te₃ is used as the n-type thermoelectric leg instead of BiTeSe alloys in this study.

5. The labeling of samples like VS-A-440-90 seems too complicated to understand, I would recommend authors simplify the labeling and highlight the most important information, like the annealing temperature or annealing time.

Reviewer #3 (Remarks to the Author):

The authors in this manuscript present their experimental and theoretical studies of high-performance micro-thermoelectric coolers that can be widely applied in microsystem thermal management. By adopting a powder direct molding method to prepare Bi₂Te₃-based thick-film thermoelectric materials and a phase-change batch transfer strategy to achieve high device filling factor, the authors developed micro-thermoelectric coolers with excellent overall cooling performance and impressive power generation performance. Their characterizations and measurements are carefully done, and the proposed method is novel and important because it provides a facile and ingenious solution that would greatly help researchers to design and fabricate thermoelectric and other micro-devices. I think this work will find an enthusiastic readership in Nature Communications, and if the authors could respond to some relatively minor points, I would recommend publication.

1. In Fig. 1d, what are the specific annealing temperature and time of the Ar annealing sample and the VS-A sample? Besides, in order to prove that the VS-A process achieved a high power factor, the authors are suggested to test the PF of the Ar annealing samples under different annealing temperatures and time.

2. In the manuscript, the authors illustrated the fast response property of the micro TEC. How about the cooling rate of commercial TECs? It is better to give a comparison in detail.

3. In the simulation section, the authors utilized both one-dimensional model and finite element analysis. Are the results of these two methods consistent?

4. I noticed that in Supplementary Fig. 8c, the ΔT_{\max} first increases and then decreases with the increase of filling factor, the authors are suggested to explain the trend.

Detailed reply to reviewers' comments

Reviewer #1:

In this manuscript, the authors prepared a thick-film micro-thermoelectric cooler through powder direct molding route and batch transfer strategy, which exhibits good cooling performance, precise temperature control and high normalized power density. The micro-TEC shows great potential in thermal management or fast cooling of microsystems. Since the design strategy is proposed for the first time to the best of our knowledge, the idea of this work is original and novel. Besides, the article is well written and the experiments are also conducted in detail. I think both the material fabrication method and the integration strategy could provide guidelines for the researchers in thermoelectric and device micromachining fields. Thus, I recommend the acceptance of the manuscript after addressing the following minor issues.

Comment 1: *In the process of fabricating thermoelectric thick films, the authors employed a homemade pre-pressing mold to control the thickness and uniformity of the films. I think it is necessary to provide a detailed description about how the pre-pressing mold is prepared in the section of "Methods".*

Author answer: Thanks very much for the referee's valuable suggestion.

In the revised manuscript, we have added the detailed preparation process of the pre-pressing mold in the "Methods" section.

For your convenience, the revised content in our manuscript is as follows:

P16/L397-408 in the revised manuscript:

The TE powders were directly prepared into thick films in several simple steps. First, a homemade pre-pressing mold was fabricated using stainless steel sheets and Al₂O₃ ceramic plates. Specifically, a stainless steel sheet with a thickness of 50-150 μm was cut into a required pattern using a femtosecond laser cutter (PHAROS-20, Amplitude Laser Group). The wavelength, pulse frequency, pulse width and power of the laser are 343 nm, 100 kHz, 290 fs and 6 W, respectively. The operation is conducted at ambient temperature. The 50 μm thick polyimide adhesive was then used to paste the patterned stainless steel sheet onto a 50 × 100 × 1 mm³ Al₂O₃ ceramic plate with an appropriate surface roughness of 200-300 nm, forming a shallow sink with a size of 30 × 30 mm² and a depth of 100-200 μm. After evenly distributing TE powders into the sink of pre-pressing mold, a punch was used to apply a pressure of 150 MPa for 3 seconds on the powders to obtain preformed thick films with a diameter of 20 mm.

Comment 2: *The authors propose a novel and interesting selective batch transfer method based on the phase transition material to achieve the integration of micro-TECs with high-density thermoelectric leg arrays. In this method, it is better to explain why rosin is chosen as the phase change material instead of other materials.*

Author answer: Thanks very much for the referee's affirmation of our work and important suggestion.

The selection of a suitable phase change material for batch transfer of thermoelectric legs involves several critical requirements: low melting point, significant adhesion changes during phase transition, and ease of cleaning. In our study, we evaluated various low melting point phase change materials, including ice, paraffin wax, polyethylene glycol, and rosin. Finally, the ethanol-soluble rosin with proper melting point of 110 °C is chosen due to its highest success rate for the clean transfer of TE legs.

The revised content in our manuscript is as follows:

P10/L250-257 in the revised manuscript:

As a result, the strength of adhesion can be significantly modulated by manipulating the intermolecular force and anchoring force, achieving batch transfer with high success rate. The selection of a suitable PCM for batch transfer of TE legs involves several critical requirements: low melting point, significant adhesion changes during phase transition, and ease of cleaning. In our study, we evaluated various low melting point phase change materials, including ice, paraffin wax, polyethylene glycol, and rosin. Finally, the ethanol-soluble rosin with proper melting point of 110 °C is chosen due to its highest success rate for the clean transfer of TE legs.

Comment 3: *In Fig. 5e, nine μ -TECs with a size of $0.7 \times 0.7 \text{ mm}^2$ were integrated on a single substrate. How is the one-time integration process of multiple devices achieved?*

Author answer: Thanks very much for the referee's helpful question.

In the revised manuscript, we have added the detailed integration process description in the "Methods" section. Besides, the digital photos of the procedures are also provided in Supplementary Fig. 13.

The revised content in our manuscript is as follows:

P12/L292-295 in the revised manuscript:

Moreover, Figure 5e demonstrates that this integration process can realize a high-density one-time integration of multiple μ -TECs on a single substrate whose integration procedures are demonstrated in Supplementary Fig. 13, providing a potential application for high-resolution array cooling or temperature sensing.

P18/L450-455 in the revised manuscript:

Similarly, the phase-change batch transfer method can also achieve the integration of multiple μ -TECs on a single substrate at once. According to the electrode positions where TE legs need to be welded, corresponding convex point stamps were prepared (Supplementary Fig. 13). After coating rosin on the convex point surfaces, P-type and N-type TE legs were picked up and alternately transferred to the PDMS substrate, followed by soldering with upper and lower electrodes.

P20 in the revised Supplementary Information:

Supplementary Figure 13. Optical images illustrating the integration process of multiple μ -TECs on a single substrate at once. First, according to the electrode positions where TE legs need to be welded, a corresponding stamp with Cu convex points (a size of $50 \times 50 \mu\text{m}^2$) is prepared on an AlN substrate. After coating rosin on the convex points, P-type and N-type TE legs are picked up and alternately transferred to the PDMS substrate, followed by soldering with upper and lower electrodes. The scale bar in the figure represents a length of 2 mm.

Comment 4: *In the section of cooling performance characterization, the authors mention that “Figure 6c illustrates that the device exhibits a fast cooling response of 4.8ms/K”. To make it easy for readers to inspect, the transient response time when changing the current should also be clearly presented in Fig. 6c.*

Author answer: Thanks very much for the referee’s precious suggestion.

To clearly present the transient response time when a current is applied, we have magnified the temperature-time curve at the instant of response. In addition, we also measured the temperature-time response of commercial TECs, demonstrating that the cooling response of our μ -TEC is much faster than the commercial mini-TEC and bulk-TEC. The magnified temperature-time curves and the comparison of TECs on transient response have been added in the inset of Fig. 6c in the revised manuscript.

The revised content in our manuscript is as follows:

P13/L321-325 in the revised manuscript:

Figure 6c illustrates that the device exhibits a fast cooling response of $4.8 \text{ ms} \cdot \text{K}^{-1}$ at 1.1 A when the cold-side temperature drops from $25 \text{ }^\circ\text{C}$ to below $0 \text{ }^\circ\text{C}$. **Notably, the cooling response of our μ -TEC is much faster than the commercial mini-TEC and bulk-TEC (the inset in Fig. 6c), which can be attributed to the higher q_{cmax} .**

P18/L462-466 in the revised manuscript:

The temperature measurement accuracy of the infrared thermal imager was calibrated by the K-type thermocouple. The transient cooling response of commercial mini-TEC and bulk-TEC (Wangu Co., Ltd., Quzhou, China) at optimal current were also characterized in the same measurement conditions.

P14 in the revised manuscript:

Fig. 6 Cooling and power generation performance of the as-integrated thick-film μ -TECs containing 50 pairs of TE legs. **a** FEM simulation of device cooling performance using COMSOL Multiphysics[®] software. **b** Cooling temperature difference as a function of input electric current. The black dots represent experimental data measured by infrared thermal imaging at 25 °C, and the red line represents the FEM results. The lower right inset shows the infrared thermal image with an applied current of 1.1 A. **c** Transient response of device cold-side temperature (T_c) when a current of 1.1 A is applied at 25 °C. **The inset shows the comparison of our μ -TEC and the commercial mini-TEC and bulk-TEC on transient response.** **d** Precise control of T_c over a wide temperature range from -5 to 25 °C. When the temperature is adjusted every 20 seconds in a step of 0.01 K, clear discernible temperature changes can be observed. **e** Cooling cycling reliability of the μ -TEC under a continuous on-off pulse current of 0.9 A, represented as the T_c as a function of cooling cycles. **f** Comparison of the as-integrated thick-film μ -TECs and reported other polycrystalline Bi_2Te_3 -based thick-film and thin-film μ -TECs on ΔT_{max} and q_{cmx} ^{5,17,43-47}. **g** Output current (dashed lines) and power (solid lines) as functions of output voltage at different temperature differences applied between the hot side and cold side of the device. **h** Comparison of the as-integrated thick-film μ -TECs and reported polycrystalline Bi_2Te_3 -based TE devices on maximum normalized voltage density (NVD) and normalized power density (NPD).

Reviewer #2:

The rapidly increased heat flux for the integrated electronics results in a daunting challenge for highly efficient heat dissipation. Designing solid-state thermoelectric cooling devices with large cooling power density is of great significance for electronic cooling/thermal management applications. Herein, Sun et al. developed thick-film thermoelectric coolers with an excellent cooling flux of 56 W cm⁻². In addition, the micro-thermoelectric devices also show high-temperature control accuracy (0.01 K) and reliability (over 30000 cooling cycles). The results are highly meaningful for the development of advanced micro-thermoelectric devices. Therefore, I would like to recommend it to be accepted by Nature Communications. My suggestions for further improving the manuscript are provided below for consideration.

Comment 1: *The cold-side temperature of the micro-TEC is obtained from the infrared thermal imager. However, since the error of the thermal imager could be large, if it is accurate to determine the temperature precision as high as 0.01 K? In addition, the Q_c and COP are based on the simulation results, if it is possible to experimentally characterize these cooling performances.*

Author answer: Thanks very much for the referee's precious suggestion.

In the experimental section, for the measurement of ΔT_{\max} , the temperature was obtained from the infrared thermal imager. While for the precise temperature control measurements, a thermistor was used to record the cold-side temperature precisely, providing a feedback for a current control module with PID to regulate the cold-side temperature with a precision as high as 0.01 K. We are so sorry for your confusion due to the previous unclear description and we have added the corresponding explanation in the revised manuscript file.

Moreover, we have sent the micro-TEC to Ferrotec Co., Ltd. for the q_c characterization. According to the test report below, the $q_{c\max}$ of 55.4 W/cm² and COP_{\max} of 0.66 at I_{\max} and 25 °C can be obtained by the equations $q_{c\max}=Q_{c\max}/A_s$ and $COP_{\max}=Q_{c\max}/(U_{\max}\cdot I_{\max})$, where A_s is the cold-side area of the micro-TEC, and U_{\max} is the measured device voltage at $T_h-T_c=0$ °C. The results are slightly smaller than the simulated values ($q_{c\max}=56.5$ W/cm² and $COP_{\max}=0.7$), which can be attributed to the effect of parasitic thermal resistance and measurement error in the experiment.

The test report is listed as follows:

 大和 TE 实验室 FERROTEC CO. LTD				 IATF16949		 ISO9001		 ISO14001																																																				
<h2>QC 测试报表</h2> Q101-2TE52-1645Z																																																												
试验名称: 外来品-北航样品分析测试 测试规格: 50对 LOT号: / 测试担当: 饶红斌 测试时间: 2024/2/4/8:58		DICE尺寸: 0.1*0.2*0.2 试验编号: 2TE5273434-3 试验担当: 李明 封胶: 未封胶																																																										
   QC测试 P1 P2 P3 QCmax  测试温度℃     1#样品 Th-Tc(℃): 12.822 6.22 0.037 3.461 1#样品 25.0   QC值(W): 2.555 3.023 3.458   U(V):   4.734                                       										QC测试	P1	P2	P3	QCmax		测试温度℃	1#样品	Th-Tc(℃):	12.822	6.22	0.037	3.461	1#样品	25.0	QC值(W):	2.555	3.023	3.458	U(V):			4.734																												
QC测试	P1	P2	P3	QCmax		测试温度℃																																																						
1#样品	Th-Tc(℃):	12.822	6.22	0.037	3.461	1#样品	25.0																																																					
	QC值(W):	2.555	3.023	3.458																																																								
	U(V):			4.734																																																								
本次QC测试条件: Th=25.003000℃; 真空度=11.380000pa. *FERROTEC保留最终解释权 * DT=0时对应的QC值即为QCmax																																																												

For your convenience, the revised content in our manuscript is as follows:

P13/L318-331 in the revised manuscript:

An experimental characterization was also performed, and q_{cmax} of 55.4 W/cm² and COP_{max} of 0.66 at I_{max} and 25 °C were obtained. The measured results are slightly smaller than the simulated values, which can be attributed to the effect of parasitic thermal resistance and measurement error in the experiment. Figure 6c illustrates that the device exhibits a fast cooling response of 4.8 ms·K⁻¹ at 1.1 A when the cold-side temperature drops from 25 °C to below 0 °C. Notably, the cooling response of our μ -TEC is much faster than the commercial mini-TEC and bulk-TEC (the inset of Fig. 6c), which can be attributed to the higher q_{cmax} . In addition, the temperature control precision and cooling cycling reliability of the device have also been demonstrated. For the temperature control measurements, a thermistor was used to record the cold-side temperature precisely, providing a feedback for a current control module with PID to regulate the cold-side temperature precisely. Figure 6d and Supplementary Figure 15 show that the cold-side temperature can be accurately controlled with a high precision of less than 0.01 K over a wide temperature range, suggesting a great potential for applications in optical chips^{1,2}.

P18/L468-472 in the revised manuscript:

For the precise temperature control measurements, a thermistor (DT103F3930, EXSENSE Sensor Technology) was used to record the cold-side temperature, providing a feedback for a current control module with PID to precisely regulate the cold-side temperature. **In addition, the experimental characterization of q_{cmax} and COP_{max} of our μ -TEC was performed in Ferrotec Co., Ltd.**

Comment 2: *The anisotropy in the thermoelectric properties of n-type Bi₂Te₃-based alloys is very noticeable, I am wondering if the thermoelectric properties of the prepared thick films are obtained from the same direction. It is understandable that the electrical properties in the out-of-plane direction and thermal conductivity in the in-plane direction are difficult to characterize for the film. Comments on the anisotropy should be added to the revised manuscript.*

Author answer: Thanks very much for the referee's insightful suggestion.

The electrical conductivity, Seebeck coefficient, and thermal conductivity shown in **Fig. 3** are characterized in the same in-plane direction of the prepared Bi₂Te₃ thick films. The specific characterization details are presented in the "Methods" section of the revised manuscript. To analyze the anisotropy of materials, the room-temperature out-of-plane Seebeck coefficient, thermal conductivity, and electrical conductivity of the prepared N-type Bi₂Te₃ thick film annealed at 440 °C for 90 min were also characterized using the portable Seebeck tester, laser flash method, and Cox-Strack method (Supplementary Fig. 5 and Note 1), respectively. These results are displayed in Supplementary Table 2. By comparing the room-temperature in-plane thermoelectric properties (**Fig. 3**), it can be found that the in-plane electrical conductivity (115170.7 S·m⁻¹) and thermal conductivity (1.371 W·m⁻¹·K⁻¹) are clearly larger than the out-of-plane electrical conductivity (103164.3 S·m⁻¹) and thermal conductivity (1.232 W·m⁻¹·K⁻¹), while the Seebeck coefficient and ZT do not exhibit significant anisotropy. We have added the comments on the anisotropy in the revised manuscript.

The revised content in our manuscript is as follows:

P7/L165-168 in the revised manuscript:

Furthermore, the temperature-dependent **in-plane TE properties** of Bi₂Te₃ thick films obtained under different annealing conditions were measured. Figure 3a shows that the TE thick films exhibit a typical metallic electrical conductivity-temperature dependence, and the RT conductivity increases with annealing temperature and time.

P7-8/L189-198 in the revised manuscript:

Consequently, the trade-off between electrical conductivity, Seebeck coefficient and thermal conductivity leads to an optimized RT ZT value of about 0.73 in the Bi₂Te₃ thick films annealed at 440 °C for 90 min (**Fig. 3f**). **To analyze the anisotropy of materials, the RT out-of-plane Seebeck coefficient, thermal conductivity, and electrical**

conductivity of the prepared N-type Bi_2Te_3 thick film annealed at 440 °C for 90 min were also characterized using the portable Seebeck tester, laser flash method, and Cox-Strack method (Supplementary Fig. 5 and Note 1), respectively. These results are displayed in Supplementary Table 2. By comparing the RT in-plane TE properties (Fig. 3), it can be found that the electrical conductivity and thermal conductivity show a clear anisotropy, while the Seebeck coefficient and ZT do not exhibit significant anisotropy.

P8 in the revised manuscript:

Fig. 3 Temperature-dependent **in-plane** TE properties of Bi_2Te_3 thick films prepared under different annealing conditions. **a** Electrical conductivity (σ). **b** Seebeck coefficient (S). **c** Power factor (PF). **d** Total thermal conductivity (κ). **e** Lattice thermal conductivity ($\kappa-\kappa_c$). **f** TE figure of merit (ZT).

Comment 3: *Since the thickness of the prepared films is only less than 200 micrometers, are the compressive strain and stress results directly obtained from the film sample? Or if the results are obtained from the micropillar sample? Details for the mechanical properties characterization should be provided.*

Author answer: Thanks very much for the referee’s important suggestion.

In the experiment, we find that it is difficult to directly measure the compressive strength of the prepared films, because its thickness is only less than 200 μm . Therefore, to obtain the mechanical properties, a thicker Bi_2Te_3 sample with a thickness of 2 mm and a diameter of 10 mm was fabricated using the powder direct molding method under the same preparation condition. As a comparison, the commercial zone melting Bi_2Te_3 sample was cut into the same size as the powder direct molding sample. Then, the compressive strain-stress curves of these Bi_2Te_3 samples were measured by an electro-mechanical universal testing machines (RGM-6300, Shenzhen Reger Instrument Co., Ltd.). In the revised manuscript, we have added these details of mechanical property characterization in the “Methods” section.

The revised content in our manuscript is as follows:

P5/L140-143 in the revised manuscript:

Indeed, Figure 2b demonstrates that the compressive strength of Bi_2Te_3 prepared by the powder direct molding method is about 3.7 times that of the commercial zone melting samples (see Methods for detailed mechanical property characterization).

P17/L428-435 in the revised manuscript:

It is difficult to directly measure the compressive strength of the prepared films, because its thickness is only less than 200 μm . Therefore, to obtain the mechanical properties, a thicker Bi_2Te_3 sample with a thickness of 2 mm and a diameter of 10 mm was fabricated using the powder direct molding method under the same preparation condition. As a comparison, the commercial zone melting Bi_2Te_3 sample was cut into the same size as the powder direct molding sample. Then, the compressive strain-stress curves of these Bi_2Te_3 samples were measured by an electro-mechanical universal testing machines (RGM-6300, Shenzhen Reger Instrument Co., Ltd.).

Comment 4: *The BiTeSe alloys have improved thermoelectric performance compared to the Bi₂Te₃. Please comment on why Bi₂Te₃ is used as the n-type thermoelectric leg instead of BiTeSe alloys in this study.*

Author answer: Thanks very much for the referee's valuable suggestion.

During the experimental process, we have also prepared $\text{Bi}_2\text{Te}_{2.7}\text{Se}_{0.3}$ via vacuum sealing and annealing method. However, the power factor of $\text{Bi}_2\text{Te}_{2.7}\text{Se}_{0.3}$ samples are lower than that of Bi_2Te_3 samples. The main reason may be the volatilization of Se element at high temperature. Thus, Bi_2Te_3 is chosen as the n-type thermoelectric leg.

Comment 5: *The labeling of samples like VS-A-440-90 seems too complicated to understand, I would recommend authors simplify the labeling and highlight the most important information, like the annealing temperature or annealing time.*

Author answer: Thanks very much for the referee's helpful suggestion.

In the revised manuscript, we have changed the labeling of samples from "VS-A-temperature-time" to "temperature-time". (Fig. 2, Fig. 3, Supplementary Fig. 3, and Supplementary Fig. 4).

The revised content in our manuscript is as follows:

P6 in the revised manuscript:

Fig. 2 Characterization of structure and composition of Bi₂Te₃ thick films prepared by the powder direct molding method. a SEM images of the surface and cross-sectional microstructure of samples annealed at 440 °C for 90 min. **b** Compressive strength of Bi₂Te₃ as a function of compression strain. The powder direct molding and commercial zone melting samples are compared. **c** Atomic percentage of Te content in Bi₂Te₃ thick films prepared at different annealing temperatures (360, 400, 440, and 450 °C) and times (70, 90, and 110 min), characterized by EDS. **d** XRD spectra of Bi₂Te₃ thick films annealed at different annealing temperatures for 90 min. **e** XRD spectra of Bi₂Te₃ thick films annealed at 440 °C for different times.

P8 in the revised manuscript:

Fig. 3 Temperature-dependent in-plane TE properties of Bi₂Te₃ thick films prepared under different annealing conditions. a Electrical conductivity (σ). **b** Seebeck coefficient (S). **c** Power factor (PF). **d** Total thermal conductivity (κ). **e** Lattice thermal conductivity ($\kappa - \kappa_e$). **f** TE figure of merit (ZT).

P10 in the revised Supplementary Information:

Supplementary Figure 3. SEM images of the surface and cross-sectional microstructure of Bi_2Te_3 thick films prepared by the powder direct molding method at different annealing temperatures (360, 400, 440, and 450 °C) and times (70, 90, and 110 min).

P11 in the revised Supplementary Information:

Supplementary Figure 4. Temperature-dependent electronic thermal properties of Bi_2Te_3 thick films prepared under different annealing conditions. a Lorenz Coefficient (L). b Electronic thermal conductivity (κ_e). The L and κ_e are calculated from the measured electrical conductivity and Seebeck coefficient.

Reviewer #3:

The authors in this manuscript present their experimental and theoretical studies of high-performance micro-thermoelectric coolers that can be widely applied in microsystem thermal management. By adopting a powder direct molding method to prepare Bi₂Te₃-based thick-film thermoelectric materials and a phase-change batch transfer strategy to achieve high device filling factor, the authors developed micro-thermoelectric coolers with excellent overall cooling performance and impressive power generation performance. Their characterizations and measurements are carefully done, and the proposed method is novel and important because it provides a facile and ingenious solution that would greatly help researchers to design and fabricate thermoelectric and other micro-devices. I think this work will find an enthusiastic readership in Nature Communications, and if the authors could respond to some relatively minor points, I would recommend publication.

Comment 1: *In Fig. 1d, what are the specific annealing temperature and time of the Ar annealing sample and the VS-A sample? Besides, in order to prove that the VS-A process achieved a high power factor, the authors are suggested to test the PF of the Ar annealing samples under different annealing temperatures and time.*

Author answer: Thanks very much for the referee's helpful suggestion.

In Fig. 1d, the Ar annealing sample was treated at 400 °C for 90 min and the VS-A sample was annealed at 440 °C for 90 min. In addition, the power factors of other Ar annealing samples were measured as exhibited in **Supplementary Fig. 2**. It can be seen that the Ar annealing sample treated at 400 °C for 90 min has a maximum *PF* of 24.08 $\mu\text{W}\cdot\text{cm}^{-1}\cdot\text{K}^{-2}$, which is lower than that of the VS-A sample (33.6 $\mu\text{W}\cdot\text{cm}^{-1}\cdot\text{K}^{-2}$). The results indicate that the thick film shows better performance after vacuum sealing and annealing process.

For your convenience, the revised content in our manuscript is as follows:

P4-5/L117-128 in the revised manuscript:

The oxygen content and TE properties of Bi₂Te₃ thick films under different annealing conditions have been compared as illustrated in Fig. 1d and Supplementary Fig. 1. Unannealed Bi₂Te₃ thick films contain approximately 4% oxygen (O) by atomic percentage due to the oxidation of TE powder in the air. Traditional annealing in a large tube furnace filled with inert gas (Ar) atmosphere leads to further oxidation of thick films by residual oxygen, resulting in a low power factor of 24.08 $\mu\text{W}\cdot\text{cm}^{-1}\cdot\text{K}^{-2}$ for Bi₂Te₃ thick films (treated at 400 °C for 90 min). In contrast, VS-A thick films (treated at 440 °C for 90 min) exhibit only slightly higher oxygen content than that of unannealed samples, indicating the effectiveness of high vacuum sealing in preventing oxidation during annealing. Moreover, confining the samples in small sealed space may reduce the volatilization of elements to a certain extent. As a result, the VS-A process can achieve a high power factor of 33.6 $\mu\text{W}\cdot\text{cm}^{-1}\cdot\text{K}^{-2}$, which is significantly higher than that of traditionally annealed thick films (**Supplementary Fig. 2**).

P9 in the revised Supplementary Information:

Supplementary Figure 2. TE properties of Bi₂Te₃ thick films prepared by traditional Ar annealing at different annealing temperatures (360, 400, 440, and 450 °C) and times (70, 90, and 110 min). a Electrical conductivity (σ). b Seebeck coefficient (S). c Power factor (PF).

Comment 2: *In the manuscript, the authors illustrated the fast response property of the micro TEC. How about the cooling rate of commercial TECs? It is better to give a comparison in detail.*

Author answer: Thanks very much for the referee's insightful suggestion.

In the revised manuscript, we have measured the temperature-time response of commercial TECs, demonstrating that the cooling response of our μ -TEC is much faster than the commercial mini-TEC and bulk-TEC. The comparison of various TECs on transient response has been added in the inset of Fig. 6c in the revised manuscript.

The revised content in our manuscript is as follows:

P13/L321-325 in the revised manuscript:

Figure 6c illustrates that the device exhibits a fast cooling response of $4.8 \text{ ms} \cdot \text{K}^{-1}$ at 1.1 A when the cold-side temperature drops from 25 °C to below 0 °C. **Notably, the cooling response of our μ -TEC is much faster than the commercial mini-TEC and bulk-TEC (the inset in Fig. 6c), which can be attributed to the higher q_{cmax} .**

P18/L462-466 in the revised manuscript:

The temperature measurement accuracy of the infrared thermal imager was calibrated by the K-type thermocouple. **The transient cooling response of commercial mini-TEC and bulk-TEC (Wangu Co., Ltd., Quzhou, China) at optimal current were also characterized in the same measurement conditions.**

P14 in the revised manuscript:

Fig. 6 Cooling and power generation performance of the as-integrated thick-film μ -TECs containing 50 pairs of TE legs. **a** FEM simulation of device cooling performance using COMSOL Multiphysics[®] software. **b** Cooling temperature difference as a function of input electric current. The black dots represent experimental data measured by infrared thermal imaging at 25 °C, and the red line represents the FEM results. The lower right inset shows the infrared thermal image with an applied current of 1.1 A. **c** Transient response of device cold-side temperature (T_c) when a current of 1.1 A is applied at 25 °C. **The inset shows the comparison of our μ -TEC and the commercial mini-TEC and bulk-TEC on transient response.** **d** Precise control of T_c over a wide temperature range from -5 to 25 °C. When the temperature is adjusted every 20 seconds in a step of 0.01 K, clear discernible temperature changes can be observed. **e** Cooling cycling reliability of the μ -TEC under a continuous on-off pulse current of 0.9 A, represented as the T_c as a function of cooling cycles. **f** Comparison of the as-integrated thick-film μ -TECs and reported other polycrystalline Bi_2Te_3 -based thick-film and thin-film μ -TECs on ΔT_{max} and q_{cmax} ^{5,17,43-47}. **g** Output current (dashed lines) and power (solid lines) as functions of output voltage at different temperature differences applied between the hot side and cold side of the device. **h** Comparison of the as-integrated thick-film μ -TECs and reported polycrystalline Bi_2Te_3 -based TE devices on maximum normalized voltage density (NVD) and normalized power density (NPD).

Comment 3: *In the simulation section, the authors utilized both one-dimensional model and finite element analysis. Are the results of these two methods consistent?*

Author answer: Thanks very much for the referee's important question.

Based on the same device parameters, we have calculated the current-dependent cooling temperature difference (ΔT) and power density (q_c) using the one-dimensional model and finite element method (FEM) simulations (Supplementary Fig. 7). The results of these two methods are consistent, demonstrating the reliability of the one-dimensional model.

The revised content in our manuscript is as follows:

P9/L216-223 in the revised manuscript:

Based on these material parameters, the current-dependent cooling temperature difference (ΔT) and power density (q_c) were calculated using the one-dimensional model and finite element method (FEM) simulations (Supplementary Fig. 7). The results of these two methods are consistent, demonstrating the reliability of the one-dimensional model. Based on the one-dimensional model, the dependence of cooling performance of TEC on the TE leg height and device filling factor was calculated with considering reasonable electrical and thermal contact resistances between the TE legs and electrodes (Supplementary Figs. 8 and 9).

P14 in the revised Supplementary Information:

Supplementary Figure 7. Comparison between the one-dimensional model and finite element method (FEM) simulations under the conditions with a hot-side temperature of 25 °C, a leg height of 100 μm , and a device filling factor of 64%. a. Cooling temperature difference (ΔT) as a function of current. b. Cooling power density (q_c) as a function of current.

Comment 4: *I noticed that in Supplementary Fig. 8c, the ΔT_{max} first increases and then decreases with the increase of filling factor, the authors are suggested to explain the trend.*

Author answer: Thanks very much for the referee’s valuable suggestion.

The non-monotonic trend of ΔT_{max} with respect to filling factor (f) can be explained by analyzing the effect of electrode resistance and substrate thermal resistance on the $\Delta T_{\text{max}}-f$ dependence (the inset in Supplementary Fig. 9). As the device filling factor increases, an increase in the parasitic thermal resistance (due to the reduction in substrate size) suppresses ΔT_{max} , while a decrease in the parasitic electrical resistance (attributable to the reduction in electrode size) enhances ΔT_{max} . Consequently, the competition between the two effects leads to the non-monotonic dependence of ΔT_{max} on f .

The revised content in our manuscript is as follows:

P6-7 in the revised Supplementary Information:

It is noteworthy that ΔT_{\max} shows a weak non-monotonic dependence on f . This interesting result can be understood by analyzing the effect of electrode resistance and substrate thermal resistance on the ΔT_{\max} - f dependence (the inset in Supplementary Fig. 9). As the device filling factor increases, an increase in the parasitic thermal resistance (due to the reduction in substrate size) suppresses ΔT_{\max} , while a decrease in the parasitic electrical resistance (attributable to the reduction in electrode size) enhances ΔT_{\max} . Consequently, the competition between the two effects leads to the non-monotonic dependence of ΔT_{\max} on f .

P16 in the revised Supplementary Information:

Supplementary Figure 9. Influence of device filling factor (f) on the cooling performance of the TEC under the conditions with a hot-side temperature of 25 °C and a leg height of 100 μm . **a, b** Cooling temperature difference (ΔT) and power density (q_c) as functions of current (I) at different f , assuming a reasonable electrical contact resistivity ($r_{ec} = 1.0 \times 10^{-10} \Omega\cdot\text{m}^2$) and thermal contact resistivity ($r_{tc} = 2 \times 10^{-6} \text{ m}^2\cdot\text{K}\cdot\text{W}^{-1}$) at the interface between the TE legs and electrodes. **c** Extracted maximum cooling temperature difference (ΔT_{\max}) as a function of f . The inset shows the effect of electrode resistance and substrate thermal resistance on the ΔT_{\max} - f dependence. The red dot line takes into account the realistic electrode conductivity (σ_{ee}) and substrate thermal conductivity (κ_s). The orange dot line neglects the electrode resistance ($\sigma_{ee}=\infty$). The blue dot line neglects the substrate thermal resistance ($\kappa_s =\infty$). **d** Extracted maximum cooling power density ($q_{c\max}$) as a function of f .

REVIEWERS' COMMENTS

Reviewer #1 (Remarks to the Author):

After the revision, the authors have dispelled all of my doubts. Now the logic of the article is clearer, the focus is more prominent, and I think it is easier for readers to understand. Now articles in this state can be received and published.

Reviewer #2 (Remarks to the Author):

The authors have addressed all my comments, please publish as is.

Reviewer #3 (Remarks to the Author):

I think the authors have responded well to the comments of the reviewers. I'd like to recommend the publication of this paper.

Detailed reply to reviewers' comments

Reviewer #1:

After the revision, the authors have dispelled all of my doubts. Now the logic of the article is clearer, the focus is more prominent, and I think it is easier for readers to understand. Now articles in this state can be received and published.

Author answer: We sincerely thank the reviewer for recognizing the improvement of our manuscript and recommending its publication in Nature Communications.

Reviewer #2:

The authors have addressed all my comments, please publish as is.

Author answer: We appreciate the reviewer's recognition of the validity of our last response, as well as the recommendation for publication.

Reviewer #3:

I think the authors have responded well to the comments of the reviewers. I'd like to recommend the publication of this paper.

Author answer: Thanks very much for your affirmation of our work. We also appreciate your kind recommendation for our paper to be published without further revisions.